# Scale invariant distribution functions in quantum systems with few degrees of freedom

**Emanuele G. Dalla Torre[1⋆]**

**1** Department of Physics and Center for Quantum Entanglement Science and Technology, Bar-Ilan University, 52900 Ramat Gan, Israel

⋆ emanuele.dalla-torre@biu.ac.il

## Abstract

Scale invariance usually occurs in extended systems where correlation functions decay algebraically in space and/or time. Here we introduce a new type of scale invariance, occurring in the distribution functions of physical observables. At equilibrium these functions decay over a typical scale set by the temperature, but they can become scale invariant in a sudden quantum quench. We exemplify this effect through the analysis of linear and non-linear quantum oscillators. We find that their distribution functions generically diverge logarithmically close to the stable points of the classical dynamics. Our study opens the possibility to address integrability and its breaking in distribution functions, with immediate applications to matter-wave interferometers.

 Check for updates

# 1 Introduction

Scale invariance is a common property of continuous phase transitions, defined through the renormalization of the space and time coordinates. Scale invariance can be used to find the universal properties of the neighboring phases, through the renormalization group (RG) method. By construction, the RG approach does not directly apply to systems described by a small number of degrees of freedom, whose dimension cannot be rescaled continuously. A fundamental question is whether these few-body systems can show a universal behavior, and how to detect it [1].

To address this question, we consider scaling transformations that act on physical observables, and we look for the invariance of their distribution functions. A trivial example is offered by constant distribution functions, which do not change when the observables are rescaled. As we will see, systems at thermal equilibrium generically belong to this universality class: under a scaling transformation of the variables, thermal fluctuations, and thus the temperature, effectively increase. In the asymptotic limit, the rescaled distributions tend to an infinite-temperature ensemble, where all possible values are equally probable, and the probability distribution is a constant. A natural direction to look for non-trivial scaling laws is offered by systems that do not thermalize, such as integrable models following a quantum quench. Several previous studies considered quenches in many-body systems and analyzed the scaling of the spatio-temporal coordinates [2]. Here, we study sudden quenches in few-body quantum oscillators and show that they give rise to probability distributions with a novel type of scale invariance.

In the context of phase transitions, it is common to define the scale invariance through the two-point correlation function $F(x_1 - x_2) \equiv \langle \phi(x_1)\phi(x_2) \rangle$, where $\phi$ is some physical property of an extended system and $x_{1/2}$ are two positions in space. A system is said to be scale invariant for large $x$ if $F$ satisfies the scaling ansatz

$$F(x) \approx \lambda^\alpha F(\lambda x), \tag{1}$$

where $\alpha$ is a critical exponent [3]. Eq. 1 is satisfied, for instance, if the correlation function decays at large distances as a power-law, $F(x) \sim x^{-\alpha}$. In this paper, we instead consider the distribution function $P$ of a physical observable $x$, and show that under appropriate conditions, $P(x)$ can be scale invariant as in Eq. 1. Specifically, the above mentioned thermal case corresponds to a situation where $P(x) = Ae^{-\alpha x^2}$, which tends to a constant for small $x$. In this paper we show that the distribution functions of quenched oscillators are generically characterized by a logarithmic divergence[4] $P(x) \approx \kappa \log(x)$, which is scale invariant because $P(\lambda x) = P(x) + \kappa \log(\lambda) \approx P(x)$, for $x \to 0$.

At an intuitive level, the scale invariance can be simply understood by considering the linearized equations of motion close to a stable point. Being linear, these equations are invariant under the scaling transformation $x \to \lambda x$, where $x$ is the distance from the stable point and $\lambda$

---

[1]One possible strategy that was discussed in the literature is to use the time axis as a scaling variable [1]. The corresponding RG approaches focus on the dynamics of individual orbitals, and help understand the transition between regular motion and chaos [2,3]. Here, we instead consider ensembles of initial conditions, and study the statistical properties of their long-time dynamics.

[2]See for example Refs. [4–19] for quantum quenches of integrable many-body systems.

[3]For a more rigorous definition of scale invariance, one may wish to consider the finiteness of the ratio between the left and right hand sides of Eq.(1), in the limit of $x \to \infty$. Note that in a scale invariant system, higher-order correlation functions are scale invariant as well. Their scale invariance is defined by extending Eq. 1 to multi-variable functions.

[4]Note that the logarithmic divergence does not pose any problem in terms of normalizability of the distribution function because $\int_0^1 dx \log(x)$ is finite.

is a constant [5]. To obtain a scale invariant ensemble, it is then sufficient to complement these equations with a scale invariant initial state, such as a particle with a fixed momentum, whose position in real space is completely uncertain. The key result of this work is that this simple phenomenon survives non linearities and is intimately related to the model's integrability.

## 2 The harmonic oscillator

We open our discussion with the analysis of an isolated harmonic oscillator $H_0 = (x^2 + p^2)/2$, where $x$ and $p$ are canonical conjugates. Here, the simplest example of a scale invariant state is offered by $|p = 0\rangle$, which satisfies $\langle x|p = 0\rangle = $ const. In a semiclassical description (which is exact for an harmonic oscillator), this state corresponds to the Wigner distribution $P(x, p) = P_0 \delta(p)$, where $P_0$ is a normalization constant [6]. Under the effects of $H_0$, this ensemble rotates in phase space: each point follows a circular trajectory around the stable point $x = p = 0$, with constant angular velocity. Thus, after time averaging, one obtains a distribution function that is inversely proportional to the circumference of a circle with radius $r = \sqrt{x^2 + p^2}$, or

$$P(x, p) = \frac{2P_0}{2\pi\sqrt{x^2 + p^2}}. \tag{2}$$

Here the factor 2 in the numerator accounts for the orbits starting from $x$ and $-x$, which contribute to the same circumference. We can now use Eq. 2 to compute the (time-averaged) marginal probability of $x$

$$P(x) = \int_{-x_0}^{x_0} dp \, P(x, p) = \frac{P_0}{\pi} \int_{-x_0}^{x_0} dp \, \frac{1}{\sqrt{x^2 + p^2}} = \frac{2P_0}{\pi} \mathrm{arsinh}\left(\frac{x_0}{|x|}\right)$$

$$\xrightarrow{x \ll x_0} -\frac{2P_0}{\pi} \log(|x|) + o(1), \tag{3}$$

where $x_0$ is an arbitrary cutoff, and $o(1)$ is a constant term that does not diverge as $|x| \to 0$. Eq. 3 shows that the distribution function of $x$ diverges logarithmically and is therefore scale invariant (see the Introduction) [7].

In this work, we show that the logarithmic divergence found in Eq. 3 is universal, because is not affected by non-linearities. This result is non-trivial because, for any finite $x$, there exists a time after which the non-linearities have a significant effect on the dynamics. The logarithmic divergence is nevertheless preserved, as long as the fixed point $x = p = 0$ is stable and the dynamics in its surroundings is characterized by invariant tori. For a scale invariant initial state, the time-averaged $P(x, p)$ is inversely proportional to the circumference of the appropriate torus, which is in turn proportional to the distance from the stable point. The integration over one variable will then generically lead to a logarithmic divergence [8].

---

[5] In this sense, the present scale invariant states can be associated with a Gaussian fixed point. At equilibrium, these fixed points offer the simplest example of scale invariant critical points. An interesting question for further studies is whether distribution functions can show non-Gaussian fixed points that are scale invariant as a consequence of non-linear terms.

[6] See Ref. [20] for an introduction to phase-space methods for quantum mechanics.

[7] This analysis can be extended to a generic harmonic oscillator with mass $m$, and natural frequency $\omega_0$: by working with normalized variables, it is straightforward to see that $P(x)$ does not depend on $m$ and $\omega_0$ (see Appendix A.1)

[8] In addition, the nonlinearities foster the observation of the scale invariant distribution function: For a non-linear system, the periods of the different trajectories are unequal, and the long-time probability distribution will generically tend to the time-averaged expression.

## 3 An integrable quantum oscillator

To exemplify this effect, we first focus on the nonlinear quantum oscillator described by the Hamiltonian

$$H = \frac{\mu}{S} S_z^2 + 2JS_x \,. \tag{4}$$

Here the spin operators satisfy $[S_x, S_y] = iS_z$ and $S_x^2 + S_y^2 + S_z^2 = S(S+1)$. Eq. 4 is named after Lipkin-Meshkov-Glick [21–23] and has a wide range of applications: It describes mean-field ferromagnets in a transverse magnetic field, as well as the two-site Bose-Hubbard model (see Appendix A.2). The equilibrium and nonequilibrium properties of Eq. 4 have been described theoretically [24–37], and realized experimentally with exciton polaritons [38,39], trapped ions [40], and ultracold atoms [41–43]. Experiments with matter-wave interferometers are particularly well suited to verify our predictions because they give natural access to the full distribution functions of the phase and number differences [44–48].

For large $S$, the Hamiltonian in Eq. 4 is well approximated by a semiclassical description [49,50], where the spin operators are substituted by two continuous variables, $n$ and $\phi$, defined by $S_z/S = n$, $S_\pm/S = \sqrt{1-n^2}\exp(\pm i\phi)/2$. The canonical variables $n$ and $\phi$ respectively correspond to the number and phase differences of the two-site Bose-Hubbard model. Under this transformation, the Hamiltonian in Eq. 4 is mapped to

$$\frac{H}{2S} = \frac{\mu}{2} n^2 + J\sqrt{1-n^2}\cos(\phi)\,. \tag{5}$$

The classical dynamics associated with this Hamiltonian has two fixed points on the line $n = 0$, respectively, at $\phi = 0$ and $\phi = \pi$. Their dynamical stability depends on the ratio between $J$ and $\mu$: for $J < |\mu|$, the system is stable only around $\phi = 0$, while for $J > |\mu|$ the system becomes stable around $\phi = \pi$ as well. This transition is associated with an equilibrium mean-field phase transition (for $\mu < 0$), or with the disappearance of macroscopic self-trapping (for $\mu > 0$) [51,52]. As we will see, this point determines a discontinuous change in the scaling properties of the distribution functions.

To achieve a scale invariant distribution function we consider the initial states $|S_z = 0\rangle$. This state corresponds to the ground state of the Hamiltonian in Eq. 4 with $J = 0$. Thus, the present dynamics is equivalent to the experimentally-relevant situation of a quantum quench in which $J$ is suddenly changed from 0 to a finite value [53–55]. In the semiclassical description of Eq. 5, this initial state is mapped to an ensemble with $n = 0$ and a uniformly distributed $\phi \in (-\pi, \pi)$, or equivalently $P(n, \phi) = \delta(n)/2\pi$. Fig. 1 shows the evolution of this ensemble, obtained by the numerical solution of the Hamilton-Jacobi equations derived from Eq. 5, for $J = 0.2\mu$. The marginal distribution $P(\phi)$ is shown in the lower panel and evolves from $P(\phi) = P_0 = 1/2\pi$ to the universal shape $P(\phi) = -(1/\pi^2)\log(\phi)$, as predicted by Eq. 3. This result confirms that the nonlinear terms present in the Hamiltonian in Eq. 5 do not affect the logarithmic divergence close to the stable point.

We now compare the above-mentioned semiclassical calculations with the exact diagonalization of the quantum Hamiltonian in Eq. 4 with $S = 1000$. In the quantum model, the logarithmic divergence can be observed in the distribution of the operator $m_y \equiv S_y/S = \sqrt{1-n^2}\sin(\phi)$, which can be approximated by $m_y \approx \phi$, in the vicinity of the stable point $n = \phi = 0$. The time-averaged distribution function of $m_y$ is defined quantum mechanically by

$$P(m_y) = \lim_{\tau \to \infty} \frac{1}{\tau} \int_0^\tau dt \, \left| \langle S_y = m_y S | \psi(t) \rangle \right|^2 , \tag{6}$$

where $|\psi(t)\rangle = e^{-i\hat{H}t}|S_z = 0\rangle$, and $H$ is the Hamiltonian in Eq. 4. As shown in Fig. 2(a), the resulting distribution function diverges logarithmically around $m_y = 0$. In actual systems, this

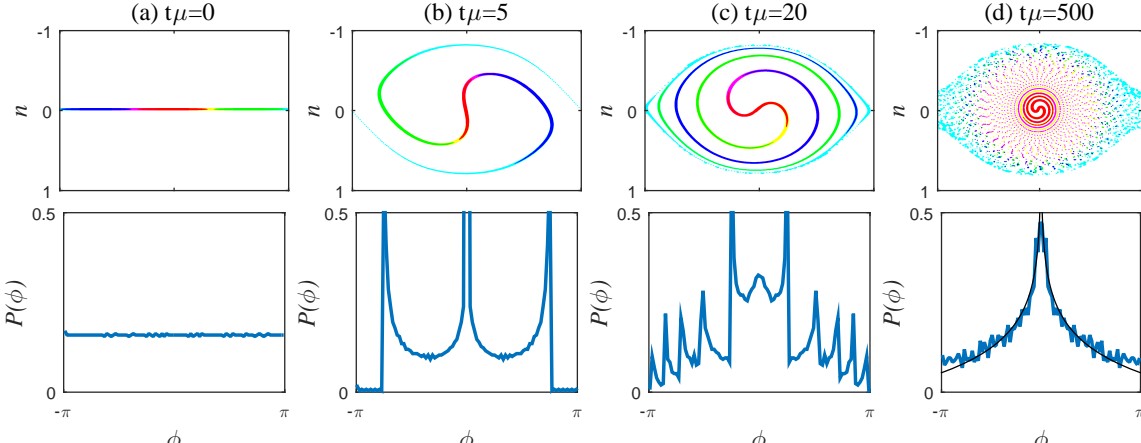

Figure 1: Upper panel: Phase-space representation of the time evolution of Eq. 5 with $J/\mu = 0.2$. Each plot represents the dynamics of 4000 points with initial conditions $n(t = 0) = 0$ and $\phi(t = 0)$ uniformly distributed between $-\pi$ and $\pi$. Each pixel is colored according to the corresponding value of $\phi(t = 0)$. (a) The initial state is $|n = 0\rangle$ and corresponds to a thin horizontal line in phase space. (b-d) Time snapshots of the evolution of the quantum ensemble. Lower panel: Time evolution of the marginal probability distribution $P(\phi(t))$. At long times $P(\phi) \approx -(1/\pi^2)\log(\phi) + 0.17$ (black line).

divergence is rounded at $1/S$, which plays the role of the infra-red cutoff of our theory (see Appendix A.3 for details). The inset of Fig. 2(a) shows that the prefactor of the logarithm suddenly jumps at $J/\mu = 1$: At this point, the number of stable points across the $m_z = 0$ line jumps from 1 to 2, leading to a doubling of the prefactor of the asymptotic distribution function [9]. A similar argument can be used to determine the universal scaling of other physical observables (see Appendix A.4).

## 4 Breaking of integrability

The logarithmic divergence of the distribution function is due to the presence of closed orbits in the vicinity of a stable point. These orbits are protected by the integrability of Eq. 4, which involves the same number of degrees of freedom $(S_x, S_y, S_z)$ as of conserved quantities $(S^2, S_z,$ and $H)$. To study the effects of integrability breaking terms, we now turn to two models where the number of degrees of freedom is larger than the number of conserved quantities: the Dicke model and the kicked rotor.

The Dicke model [56] is a canonical model of quantum optics. It describes the interaction between a quantized cavity mode $(a)$ and a large ensemble of spins (or, equivalently, a single large spin $S$). In the thermodynamic limit of $S \to \infty$, the Dicke model undergoes a phase transition from a normal to a super-radiant phase [57,58], at a critical value of the cavity-spin coupling, $\lambda = \lambda_c$. This transition was throughly described both at equilibrium and out-of-

---

[9]A closer inspection of Fig. 2 shows that for $J < \mu$, $P(m_y)$ shows a cusp at finite $m_y$. This cusp is associated with two additional stable fixed points at $S_z \neq 0$, which correspond to the two ferromagnetic equilibrium states. The presence of these stable points is at the origin of the macroscopic quantum self-trapping effect. As approaching $J = \mu$, the cusp shifts to smaller $m_y$ and, for $J > \mu$, it joins the divergence at $m_y = 0$, doubling the prefactor of the logarithm. A similar behavior can be obtained by the numerical solution of the semiclassical equations of motion associated with Eq. 5.

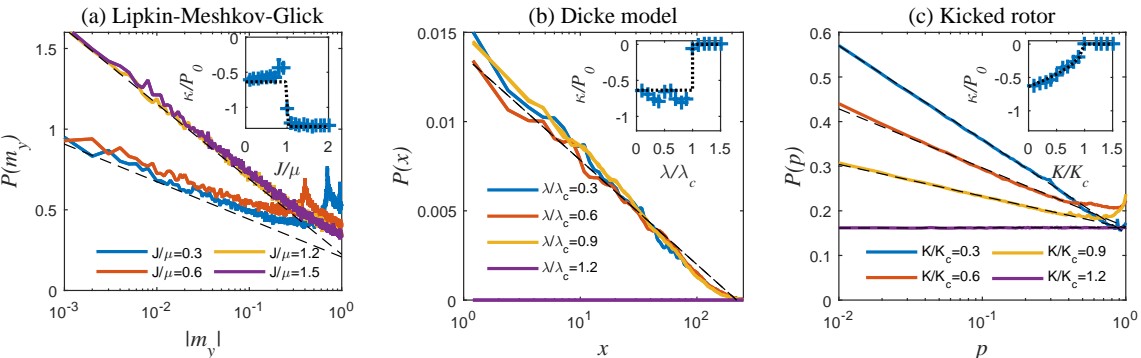

Figure 2: Time-averaged probability distribution function of physical observables for three different models: numerics (continuous curves) and logarithmic fits (dashed lines). In all three model the distribution function diverges logarithmically close to the fixed points of the classical dynamics. Inset: The prefactor of the logarithmic divergence, $\kappa$, shows a non-analytic behavior at phase transitions and at the onset of chaos.

equilibrium, and recently observed in cavity-QED [10]. The Dicke model has the same number of conserved quantities as the model defined in Eq. 4, but one additional degree of freedom. As a consequence, the Dicke model can give rise to a chaotic motion, whose onset occurs in the close vicinity of the phase transition [60].

We numerically simulate the Dicke model using the semiclassical equations of motion derived in Ref. [60], which are valid for $S \gg 1$ (see Appendix A.5). Our initial state corresponds to a pure state where $\langle S_z \rangle = -S/2$, and the photon is largely squeezed, to mimic a scale invariant state. At long times, the probability distribution of the squeezed quadrature diverges logarithmically (See Fig. 2(b)). The prefactor of the logarithm is constant for all $\lambda < \lambda_c$, and equals that in Eq. 3. At the critical coupling $\lambda_c$, the system becomes chaotic and tends to thermalize: correspondingly, the logarithmic divergence suddenly disappears (see the inset of Fig. 2(b)).

We next move to a canonical model used to describe the transition between regular and chaotic dynamics, the kicked rotor (see Appendix A.6). This model has a fixed point at $x = p = 0$, whose vicinity becomes chaotic at a critical value of the kick strength $K_c = 4$. In Fig. 2(c), we show the long-time distribution obtained from an initial ensemble with a uniformly distributed momentum $p \sim U(0, 2\pi)$ and a constant position $x = 0$. We observe that the distribution function of $p$ develops a logarithmic divergence close to $p = 0$. Interestingly, we find that the prefactor is not constant, but follows the empirical law $\kappa = (-2P_0/\pi)\sqrt{1 - K/K_c}$. This curve is non-analytic at $K_c$, at the onset of chaos, where the logarithmic divergence is washed out. These findings strengthen the relation between the integrability and the logarithmic divergence of the probability distribution [11].

## 5 Beyond Hamiltonian systems: dissipation

We now turn to study the effects of dissipation, relevant to the experimental realization with

---

[10]See Ref. [59] for an introduction to the superradiant transition of the Dicke model

[11]Note that the present semi-classical analysis does not take into account the dynamical localization due to quantum coherence [61]. The consequences of this effect on the logarithmic divergence requires further investigation.

matter-wave interferometers [53, 62]. We model this effect by

$$\frac{d\phi}{dt} = n - \Gamma \frac{n}{\sqrt{1-n^2}} \cos(\phi),$$
(7)

$$\frac{dn}{dt} = -\Gamma \sqrt{1-n^2} \sin(\phi) - 2\eta n,$$
(8)

where $\eta$ is the dissipation rate [12]. In the limit of $\eta \to 0$ these equations of motion are equivalent to the Hamilton-Jacobi equations associated to Eq. 4. The dissipative term is invariant under the scaling transformation $\phi \to \lambda\phi, n \to \lambda n$: As demonstrated by the numerical calculations of Fig. 3(a) (for $J/\mu = 0.2$, $\eta/\mu = 0.1$) the distribution of $\phi$ is still logarithmically divergent, although the prefactor becomes time dependent.

To understand this behavior, we go back to the phase-space picture, where each point follows a spiral motion (inset of Fig. 3(b)). Close to the stable point, the motion is described by a damped harmonic oscillator, whose solution gives $\phi(t) = \phi_0 e^{-\eta t} \cos(\omega t)$. As a consequence, the phase-space density grows as $e^{\eta t}$ and the time-averaged distribution is given by

$$P_\tau(\phi, p) = \frac{1}{\tau} \int_0^\tau dt \, \frac{2P_0/\pi e^{\eta t}}{\sqrt{\phi^2 + n^2}} = \left(\frac{e^{\eta\tau}-1}{\eta\tau}\right) \frac{2P_0/\pi}{\sqrt{x^2+p^2}},$$

$$\text{and} \quad P_\tau(\phi) \approx -\frac{2P_0}{\pi} \left(\frac{e^{\eta\tau}-1}{\eta\tau}\right) \log(|\phi|).$$
(9)

As shown in Fig. 3(b), this expression is in quantitative agreement with the numerical solution of the full non-linear model.

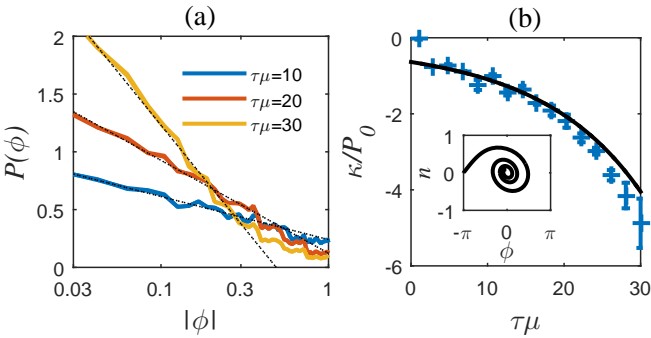

Figure 3: (a) Time-averaged probability distribution of $\phi$ in the presence of dissipation, for different waiting times $\tau$. (b) Prefactor of the logarithm, obtained by a numerical fit of the form $P_\tau(\phi) = \kappa(\tau)\log(|\phi|)$ (crosses) and by the analytical expression Eq.9 (black line). Inset: phase-space trajectory in the presence of dissipation.

## 6 Conclusion: Towards a full scaling theory

A logarithmic divergence is invariant under the scaling transformation $x \to \lambda x$, and this property can be used to address the effect of generic perturbations. Under the scaling transformation, all non-linear terms appearing in the equations of motion tend to zero ("irrelevant").

---

[12]Note that our dissipative term differs from the expression used in Ref. [62], where a force proportional to $-\eta(d\phi/dt)$ was considered. Our *linear* term has a phenomenologically similar effect, but simplifies the calculation of the correspondent fluctuating forces.

These terms do not affect the logarithmic divergence of the distribution functions (Figs. 1 and 2(a-b)). Linear perturbations are invariant under the scaling transformation ("marginal"): These terms modify the prefactor of the logarithmic divergence, and eventually lead to its disappearance (Figs. 2(c) and 3).

Finally, if a term does not depend on $x$, it effectively grows under the scaling transformation, and destroys the logarithmic divergence ("relevant"). A natural example is offered by the random forces associated with a coupling to a thermal bath. These forces generically drive the system towards an equilibrium distribution function, of the form $P_{\mathrm{eq}}(\phi) = P_0 \exp(-E(\phi)/T)$, where the $E(\phi)$ is the energy. This expression is analytical around $\phi = 0$, indicating that $P(\phi)$ does not diverge. To study this effect numerically, we consider Eq. 8 with an additional stochastic force $f(t)$. According to the fluctuation-dissipation theorem, this force satisfies $\langle f(t) \rangle = 0$, and $\langle f(t)f(t') \rangle = 4\eta T \delta(t-t')$, where $T$ is the temperature of the bath. As shown in Fig. 4 (at temperature $T = 0.1$), the system flows towards a thermal distributions, and the logarithmic divergence is destroyed.

The logarithmic divergence of the distribution function is therefore a clear indicator of the absence of thermalization in quenched oscillators. Our scaling theory can be used to analyze the effect of generic perturbations (see Appendix A.7). This approach shows a possible way to generalize our findings to many-body systems: the Lipkin-Meshkov-Glick and Dicke models are exact mean-field solutions of interacting systems with infinite-range interactions. By considering the perturbations induced by a finite-range, it will be possible to study the crossover to extended many-body systems. Finally, by including the effects of disorder, one can attempt to describe the non-Gaussian distribution functions that were recently found in quantum quenches of many-body-localized systems [63].

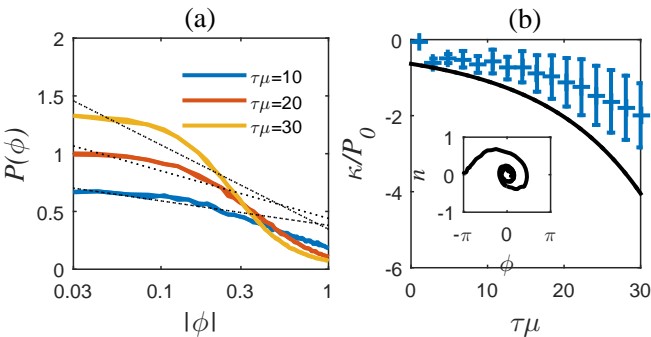

Figure 4: Same as Fig. 3, in the presence of a thermal noise at temperature $T = 0.1$. The system thermalizes and the logarithmic divergence of the distribution function is destroyed.

# Acknowledgements

We thank Baruch Barzel, Eugene Demler, Jonathan Karp, Marine Pigneur, Shoumi Roy, Angelo Russomanno, Jörg Schmiedmayer, Thomas Schweigler for many useful discussions. This work was supported by the Israeli Science Foundation Grant No. 1542/14.

# A  Appendix

## A.1  Harmonic oscillator with non-unit mass and frequency

In the main text we considered an harmonic oscillator with natural frequency $\omega_0 = 1$, and mass $m = 1$, whose phase-space orbits are circles. Let us now consider an harmonic oscillator of the form $H = (p^2/m + m\omega_0^2 x^2)/2$. Its equations of motion are given by

$$\frac{dx}{dt} = \frac{p}{m} , \quad \frac{dp}{dt} = -m\omega_0^2 x . \tag{10}$$

For convenience, we now introduce the rescaled variables $\hat{x} = \sqrt{m\omega_0}x$ and $\hat{p} = p/\sqrt{m\omega_0}$, whose equations of motion are

$$\frac{d\hat{x}}{dt} = -\omega_0\hat{p} , \quad \frac{d\hat{p}}{dt} = -\omega_0\hat{x} . \tag{11}$$

If we rescale the time to $\hat{t} = t\omega_0$, we are back to the case discussed in the main text. Thus, using Eq. 3, we find that the time-averaged distribution function of $\hat{x}$ (for small $\hat{x}$) is

$$P(\hat{x}) \approx -\frac{2\hat{P}_0}{\pi}\log(\hat{x}). \tag{12}$$

Here $\hat{P}_0$ is determined by the initial conditions, given by $P(\hat{x}, \hat{p}) = P(\sqrt{m\omega_0}\hat{x}, \hat{p}/\sqrt{m\omega_0}) = P(x, p) = P_0\delta(\sqrt{m\omega_0}\hat{p}) = P_0\delta(\hat{p})/\sqrt{m\omega_0}$, or equivalently $\hat{P}_0 = P_0/\sqrt{m\omega_0}$. Using this relation, we finally obtain

$$P(x) = P\left(\frac{\hat{x}}{\sqrt{m\omega_0}}\right) = \sqrt{m\omega_0}P(\hat{x}) \approx -\frac{2P_0}{\pi}\log(x). \tag{13}$$

Importantly, Eq. 13 does not depend on $m$ or $\omega_0$, giving a first hint about the universality of this result.

## A.2  Two-site Bose-Hubbard model

The two-site Hubbard model is described by the Hamiltonian

$$H = \frac{\mu}{N}\sum_{i=1,2}\left(\psi_i^\dagger\psi_i - \frac{N}{2}\right)^2 + J(\psi_1^\dagger\psi_2 + h.c.), \tag{14}$$

where $\mu$ is the chemical potential, and $J$ the tunneling element. Because the model commutes with the total number of particles, we restrict ourself to the subspace with a fixed $N = \psi_1^\dagger\psi_1 + \psi_2^\dagger\psi_2$.

The Hamiltonian in Eq. 14 is conveniently described in terms of $N$ spin-1/2 variables, $\vec{\sigma}_i$, whose $z$ component describes the site occupied by the $i^{\text{th}}$ particle [64,65]. This mapping is formally achieved through the Schwinger boson representation of spin operators $S_\alpha = 1/2\sum_{i,j=1,2}\hat{\psi}_i^\dagger\sigma_\alpha^{i,j}\psi_j$, where $\alpha = x,y,z$ and $\sigma_\alpha$ are Pauli matrices. By introducing the total spin operator $\vec{S} = \sum_{i=1}^N\vec{\sigma}_i$, one can exactly map Eq. 14 to the Lipkin-Meshkov-Glick model, Eq. 4 of the main text, with $S = N/2$.

## A.3 Finite size scaling

Our derivation of a scale invariant distribution functions relies on a semiclassical description of a quantum model. Specifically, the analysis of the Lipkin-Meshkov-Glick model of Eq. 4 referred to the limit $S \to \infty$, where the quantum spin becomes a semiclassical rotor. In this appendix we consider the effects of a finite $S$. For this purpose, we study the steady-state distribution functions of the model for different values of $S$. As shown in Fig. 4, the logarithmic divergence is already evident for $S = 250$. Because the minimal value of $m_y = S_y/S$ is $1/S$, the distribution function is terminated at this value. As $S$ increases, the cutoff becomes smaller, and the logarithmic divergence more pronounced. Thus, a finite $S$ has a similar role to the infra-red (IR) cutoff of a scale invariant theory, which is usually determined by the finite size of the system.

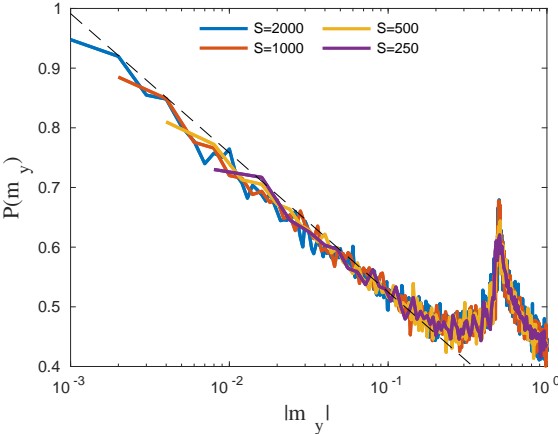

Figure 5: Steady-state distribution function of $m_y = S_y/S \approx \phi$, Eq. 6, for the Lipkin-Meshkov-Glick model, Eq. 4 with $J/\mu = 0.5$ and different values of the total spin S. The constant of motion $S$ plays the role of an IR cutoff for the scale invariant distribution function.

## A.4 Other observables

In the main text, we focused on the probability function of the variables $\phi$ and $n$ and we showed that they diverge logarithmically around the stable fixed point $n = \phi = 0$. The distribution function of other physical observables can be directly computed from $P(n, \phi)$. For instance, let us consider the operator $m_x = S_x/S = \sqrt{1-n^2}\cos(\phi)$. Close to the stable point $n = \phi = 0$, this quantity can be approximated as $m^x \approx 1 - (n^2 + \phi^2)/2$. Following the same arguments as in Sec. A.1 we obtain

$$P(1-m_x) \approx P(n^2 + \phi^2) = \frac{P_0}{\pi} \frac{1}{\sqrt{n^2 + \phi^2}} = \frac{P_0}{\pi\sqrt{1-m_x}} \ . \tag{15}$$

This result is numerically confirmed in Fig. 6.

## A.5 Dicke model

The Hamiltonian of the Dicke model [56] is

$$H = \hbar\omega_0 S_z + \hbar\omega a^\dagger a + \frac{\lambda}{\sqrt{2S}}(a + a^\dagger)(S^+ + S^-) \ . \tag{16}$$

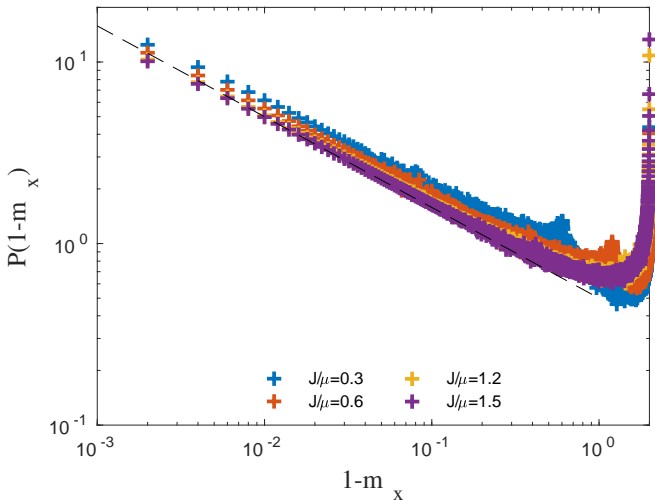

Figure 6: Steady-state distribution function of $1-m_x = 1-S_x/S \approx \phi^2$, for the Lipkin-Meshkov-Glick model, Eq. 4. The probability function of this quantity diverges as $\sqrt{1-m_x}$ (dashed line) on both sides of the transition.

Here $S$ is a spin operator (as in the main text) and $a$ is a canonical bosonic operator satisfying $[a, a^\dagger] = 1$. In the limit of $S \to \infty$, this model undergoes a phase transition [57, 58] at $\lambda_c = \sqrt{\omega_0 \omega}/2$.

For large $S$, the Dicke model in Eq. 16 is well approximated by the semiclassical Hamiltonian (Eq. 65 of Ref. [60])

$$
\begin{aligned}
H_{sc} = -j\omega_0 \quad &+ \quad \frac{1}{2}\left(\omega^2 x^2 + p_x^2 - \omega + \omega_0^2 y^2 + p_y^2 - \omega_0\right) \\
&+ \quad 2\lambda\sqrt{\omega\omega_0}\, xy\sqrt{1 - \frac{\omega_0^2 y^2 + p_y^2 - \omega_0}{4j\omega_0}},
\end{aligned}
\tag{17}
$$

were $x$, $p_x$, $y$, and $p_y$ are two pairs of canonical coordinates (associated with the two quadratures of of the cavity boson, and of the spin, respectively).

The correspondent equations of motion are (Eqs. 68-69 of Ref. [60])

$$
\begin{aligned}
\dot{x} &= p_x, \\
\dot{y} &= p_y\left(1 - \frac{\lambda}{2j}\sqrt{\frac{\omega}{\omega_0}}\frac{xy}{\sqrt{1-\eta}}\right), \\
\dot{p_x} &= -\omega^2 x - 2\lambda\sqrt{\omega\omega_0}\,y\sqrt{1-\eta}, \\
\dot{p_y} &= -\omega_0^2 y - 2\lambda\sqrt{\omega\omega_0}\,x\sqrt{1-\eta}\left(1 - \frac{\omega_0 y^2}{4j(1-\eta)}\right),
\end{aligned}
\tag{18}
$$

where

$$
\eta = \frac{1}{4j\omega_0}\left(\omega_0^2 y^2 + p_y^2 - \omega_0\right).
\tag{19}
$$

This model shows a transition between regular and chaotic motion at $\lambda \approx \lambda_c$.

In our numerical calculations, we considered $S = 10^6$. The initial state of the spin was chosen to represent the quantum state $|S_z = -S\rangle$ (which corresponds to the ground state of the

model for $\lambda = 0$). In the semiclassical picture, this state is represented by a Wigner distribution in which $y$ and $p_y$ are extracted from Gaussian ensembles with zero average and variances $1/(2\omega_0)$ and $\omega_0/2$, respectively. The state of the boson was chosen to represent a vacuum squeezed state with $\langle x \rangle = \langle p \rangle = 0$, $\langle x^2 \rangle = 10^6/4$ and $\langle p^2 \rangle = 10^{-6}$, satisfying the minimal uncertainty relation between canonical variables. The model's parameters are chosen such that the frequency of the $x$ and $y$ oscillators are incommensurate: $\omega_0 = 1/\sqrt{2}$ and $\omega = \sqrt{3}$. We observed empirically that the case $\omega = \omega_0$ gives rise to a distinct behavior, which requires further investigation. The equations of motion were solved using the Euler method with time-step discretization of $dt = 0.01$, and the distribution functions were averaged over times up to $t = 100$.

### A.6  Kicked rotor and Chirikov standard map

The Hamiltonian of the kicked rotor is (See Ref. [66] and references therein)

$$H(t) = \frac{1}{2}p^2 - K\cos(x)\sum_n \delta(t - nT),$$ 
(20)

where $\delta$ is the Kronecker delta function. Note that in previous literature, the model is often defined with an opposite sign of $K$, or equivalently after the transformation $x \to x + \pi$.

The stroboscopic dynamics of the model (i.e. the evolution of the system after a discrete number of time periods) is governed by the Chirikov standard map

$$p_{n+1} = p_n - K\sin(x_n),$$ 
(21)

$$x_{n+1} = x_n + p_{n+1}.$$ 
(22)

Due to the periodicity of the model, it is then common to define the dynamics on a torus, where $x$ and $p$ are restricted to the interval $(0, 2\pi)$.

The dynamics of the model in Eq. 22 is characterized by three distinct regimes: For $K < K_c \approx 0.9716$ the model is localized between invariant tori (i.e. $p$ does not grow with time); For $K_c < K < 4$ the model has a mixed phase space, where the dynamics is diffusive for most some conditions, and localized in vicinity of the stabel point $x = p = 0$; For $K > 4$ the region around the stable point becomes chaotic.

### A.7  Extended Lipkin-Meshkov-Glick model

In this section we explain how to apply the scaling analysis to predict the effect of non-linear terms on the logarithmic divergence. For this task, we consider the a generalization of Eq. 4, which includes two additional terms

$$H = \frac{\mu}{S}S_z^2 + 2JS_x + \alpha S_z + \frac{\beta}{S}S_x^2.$$ 
(23)

Within the semiclassical approach, the first term, $S_z = n$ enters into the equations of motion of $d\phi/dt$ as a constant term. This term grows under scaling and destroys the logarithmic divergence. In contrast, $S_x^2 = (1 - n^2)\cos^2(\phi)$ is a non-linear perturbation and does not affect the logarithmic divergence. These predictions are verified numerically in Fig. 7, where we consider the initial state $|S_z = 0\rangle$ with S=1000, evolve it in time with the Hamiltonian of Eq. 23, and compute the (time averaged) distribution probabilities of the operator $m_y = S_y/S$. As predicted by the scaling analysis, the coupling $\alpha$ destroys the logarithmic divergence, while $\beta$ leaves it unchanged.

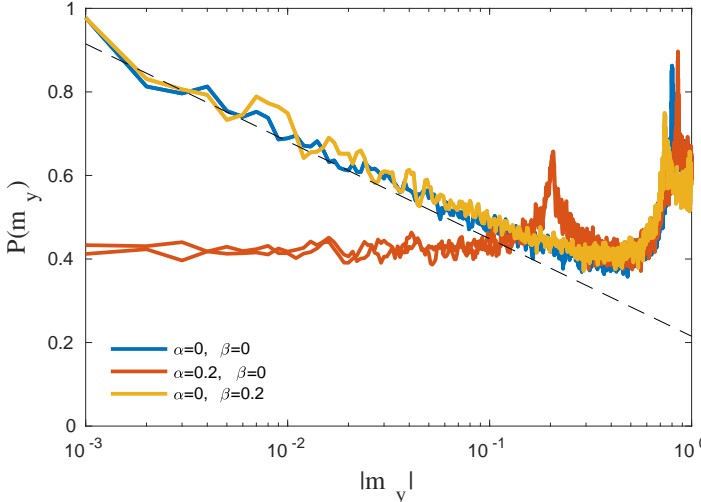

Figure 7: Steady-state distribution function of $m_y = S_y/S \approx \phi$, Eq. 6, for the generalized Lipkin-Meshkov-Glick model, Eq. 23. The coupling $\alpha S_z$ is relevant and destroys the logarithmic divergence, while $\beta S_x^2$ is irrelevant and does not affect it.

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
