# Peer review of "Scale invariant distribution functions in quantum systems with few degrees of freedom"

_SciPost Physics, doi:SciPost Phys. 5, 023 (2018)_

## Round 3 · Referee Report · Anonymous (Referee 1) · 2018-3-10

Strengths

  1. Establishes an "order parameter" for detecting integrability in quantum systems with few degrees of freedom via the behavior of the Wigner probability distribution functions.
  2. Creative and original idea.

Weaknesses

  1. Interpretation in terms of scale invariance: unclear if not misleading, or necessary for the understanding.

Report

The paper by Dalla Torre points out a new criterion for detecting integrability in quantum systems with few degrees of freedom via the behavior of the Wigner probability distribution functions near fixed points of the Hamiltonian dynamics. More precisely, it is elaborated that the spatial probability density exhibits a logarithmic divergence in the vicinity of such fixed point. This is divergence is directly related to the fact that by construction, in the vicinity of a fixed point, the dynamics can be linearized. Therefore, the result applies to intrinsically linear systems, but also to integrable non-linear systems, whose near fixed point dynamics is likewise characterized by closed periodic orbits indistinguishable from a linear one. It is demonstrated that the logarithmic divergence is absent in non-integrable systems, and that the coefficient of the log can be used as an order parameter for the presence of integrability. Moreover, it is shown that systems exposed to thermal noise do not show the divergence.

From my point of view, this is an interesting work, and fun to read. The insight that integrability can be detected based on properties of the distribution function appears new to me, and should be a realistic object to measure in experiments as argued by the author. It is clearly written and I have no doubts that the results are correct. I would recommend it for publication after the following points of criticism have been addressed by the author.

Requested changes

  1. The need for the concept of scale invariance is unclear to me. Is it really necessary to understand what is going on? Or could it be even misleading (even in that case, this would not degrade the value of the paper, but the presentation should be adapted)? To this end, I recapitulate the point: Near a fixed point, by definition any dynamics can be linearized: No need to argue with scale invariance. For the initial state, it appears to me that what is necessary is to have some probability already in the initial state near the fixed point, which is automatically fulfilled if the author's notion of scale invariance is used. Conversely: Can the author prove that for other, non-scale invariant initial states, the phenomenon is absent? Take e.g. a broad Gaussian in x and a narrow one in p instead of \delta (p): This violates scale invariance, but does it kill the log? Moreover, from my understanding also non-integrable systems could be linearized around their fixed points and initialized scale invariantly. But the scale invariant log is absent. So, it does not to be a concept that causally explains what is going on in the system. If so, the paper should be rewritten accordingly. Finally, in a more conventional understanding of scale invariance, one would expect it only close to critical points or in phases with soft modes. None of these circumstances occur here. Also, momenta would be scaled inversely to positions, again not possible here.
  2. Just technically, in which sense is the log scale invariant? Should it be \log | x| /x_0 in Eq. (2)? Do I have to rescale x and x_0? A log is not a homogeneous function (f (\lambda x ) = \lambda^a f(x), a some exponent) and therefore would not be connected to scale invariance usually.
  3. Is there a more physical explanation of why the log is absent in the chaotic system, ideally beyond scale invariance? There should still be closed orbits, just they should be very sensitive to small perturbations?
  4. I do not get the point of the discussion of constant drive terms in the dynamical equations. By definition, such terms are absent at a fixed point.

Some suggestions on wording: 5. "few-body quantum systems" could be misleading to people working in ultracold atoms. This could suggest a connection to few atom systems in spatial continuum, e.g. atomic collisions or Efimov physics. "quantum systems with few degrees of freedom" would be more appropriate, from my point of view. 6. The notions "stable point", "scale transformation", "scale invariant" should really be defined more clearly in the initial sections, maybe using more examples (if the concept is not abolished, see above). In particular, the notion of scale invariant state was completely opaque to me. 7. typos: p. 4 "with with" -> "with", p. 6 "logarithmically divergence" -> "logarithmically divergent" 8. What does "n(t = 0) = 2000" mean in the caption of Fig. 1? It seems n(t=0) =0 form the figure.

  • validity: high
  • significance: good
  • originality: high
  • clarity: high
  • formatting: excellent
  • grammar: good

Author:  Emanuele Dalla Torre  on 2018-07-18  [id 297]

(in reply to Report 1 on 2018-03-10)
Category:
answer to question

I thank the Referee for carefully reading my manuscript and for her/his useful comments.

  1. As the Referee correctly mentions, the goal of this paper is to find a clear criterion to identify integrability by using distribution functions. I would like to first answer her/his comment about non-integrable models. The Referee correctly states that their dynamics can be linearized. However, as I now mention in the paper, for any finite $x$, there exists a time after which the non-linearities have a significant effect. In an integrable model, the logarithmic divergence is nevertheless preserved, as long as the stability of the $x=p=0$ point is preserved and the dynamics in its surroundings is characterized by invariant tori. Thus, the logarithmic divergence is a clear indication of integrability.

The Referee correctly points out that my models do not support soft modes and/or the rescaling of coordinates and momenta. This is indeed the key novelty with respect to the conventional idea of scale invariance. I now clarify this point in the third paragraph of the introduction [Our definition…], where I explain what I mean by scale invariance, and how this differs from the usual case. This paragraph also clarifies the relation between the logarithmic divergence and the scale invariance. In short, the scale invariance is used here to classify different types of distribution functions and distinguish the thermal case from the non-thermal case of quantum quenches in intergrable models.

  1. I now define the concept of scale invariance and explain why a logarithm satisfies this definition in the third paragraph of the introduction [Our definition…].

  2. In a chaotic system, the orbits are not closed, but “randomly” jump from one point in phase space to another. As a consequence, any initial probability distribution spreads evenly in phase space. This effect is best exemplified by the phase space portraits of the kicked rotor - see for example https://commons.wikimedia.org/wiki/File:Kicked_Rotor_Phase_Portrait.png

  3. My discussion deals with random forces that average to zero and do not affect the stability of the fixed point. This is now clarified in the text: "A natural example is offered by the random forces associated with a coupling to a thermal bath."

  4. I thank the Referee for the suggestion and updated the title accordingly.

  5. Following the Referee’s suggestion, I now added a paragraph to the introduction, where I define the concept scale invariance in the usual sense, and relate it to my new work.

  6. Fixed

  7. I thank the Referee for spotting this severe typo, which was now fixed.

---

## Round 3 · Referee Report · Anonymous (Referee 2) · 2018-3-15

Strengths

1)Interesting idea introduced (scale invariance in distributions) 2) tested with examples of increasing complexity 3)clarity of exposition 4)different classes of models considered, diversified approaches (numerics/analytics)

Weaknesses

1)Introduction section and general framework for the work could be improved
2)Discussion of generality of the results (beyond simple model considered, choice of observable) could be improved

Report

This work introduces the concept of scale invariant distribution of physical observables and discuss it through examples of increasing complexity, moving from an harmonic oscillator, to a large spin model up to a Dicke and Kicked rotor models. The main result for all the above cases is a logarithmic scaling form for the marginal probability distribution of a certain observable, a result which is obtained both analytically and numerically.
While the results obtained are interesting and clearly exposed, I think the current manuscript lacks for what concerns the introduction and the discussion of the generality of the results. Also, some further analysis of the models considered in the paper could be useful.
Specifically:
1)Introduction
The author could put the study of scale invariance in distribution functions in a broader perspective and explain why this is interesting/new (and for example beyond what typically studied in the context of scale invariance in critical phenomena, if that's the case). Also the focus of the work could be clarified, the author mentions few-body systems but then consider more many body like implementations (Large Spin, Dicke,...). The discussion about thermal case under scaling could also benefit from some clarification, as it is a bit disconnected from the rest.

2)Role of Quantum Fluctuations/Thermodynamic Limit
My major concern/question about this work is that the author focuses on models (beside the harmonic oscillator) in which the thermodynamic limit coincides with a semiclassical limit being exact (large spin, Dicke,etc..). In this respect it is unclear whether the scale invariant distribution would survive say in a model with short range interactions (in which the thermodynamic limit is genuinely quantum). Also, when the author talks about integrability it is not clear whether it refers to integrability of the associated classical dynamics or not.
Finally, it would be interesting if the author could make a more systematic study of the finite size dependence of the numerical results (say figure 1-3, for different values of S). For example, the large spin model is studied analytically and numerically for S=2000, which is already pretty high, but it would be nice to see what happens to the log-singularity when the model is genuinely quantum and far away from the semi-classical regime.

3)Role of observables
The author could also discuss how the choice of the observable affects the scaling. For example, in the large spin model what would happen to the distribution of Sx, rather than Sy?

Requested changes

See above (report) for more details 1) Improve the introduction (see above) 2) discuss role of quantum fluctuations/thermodynamic limit and what happens for finite range interactions 3) study finite-S dependence of distribution function 4)comment on the role of observable

minor change: Add the value of the parameter S in caption of figure 1 (and similarly for other figures, for example figure 2 and Dicke model)

  • validity: high
  • significance: good
  • originality: good
  • clarity: high
  • formatting: excellent
  • grammar: excellent

Author:  Emanuele Dalla Torre  on 2018-07-18  [id 296]

(in reply to Report 2 on 2018-03-15)

I thank the Referee for carefully reading manuscript and for her/his useful suggestions.

  1. Following the Referee's suggestion, I added a paragraph to the introduction ["Our definition…"], where I explain the difference between the new type of scale invariance and the common one. Furthermore, this paragraph explains the connection between the scale invariance of thermal states and of the new non-equilibrium states is now explained in the introduction. In addition, the title has been changed to specify that we deal with systems with few degree of freedom, rather than true few body problems.

  2. I agree with the Referee that the extension of these concepts to many-body situations is highly desirable. However, this will require further studies that go beyond the scope of the present paper.

  3. The spin size S determines the minimal value of $S_y$ and thus terminates the logarithmic divergence, playing the role of an infrared cutoff. This point is now discussed in appendix A.3 (see the new figure 5)

  4. . The distribution function of other physical observables is now studied in the appendix A.4 and in the new Figure 6.

Author:  Emanuele Dalla Torre  on 2018-07-18  [id 295]

(in reply to Report 2 on 2018-03-15)

I thank the Referee for carefully reading my manuscript and for her/his useful comments.

  1. As the Referee correctly mentions, the goal of this paper is to find a clear criterion to identify integrability by using distribution functions. I would like to first answer her/his comment about non-integrable models. The Referee correctly states that their dynamics can be linearized. However, as I now mention in the paper, for any finite $x$, there exists a time after which the non-linearities have a significant effect. In an integrable model, the logarithmic divergence is nevertheless preserved, as long as the stability of the $x=p=0$ point is preserved and the dynamics in its surroundings is characterized by invariant tori. Thus, the logarithmic divergence is a clear indication of integrability.

The Referee correctly points out that my models do not support soft modes and/or the rescaling of coordinates and momenta. This is indeed the key novelty with respect to the conventional idea of scale invariance. I now clarify this point in the third paragraph of the introduction [Our definition…], where I explain what I mean by scale invariance, and how this differs from the usual case. This paragraph also clarifies the relation between the logarithmic divergence and the scale invariance. In short, the scale invariance is used here to classify different types of distribution functions and distinguish the thermal case from the non-thermal case of quantum quenches in intergrable models.

  1. I now define the concept of scale invariance and explain why a logarithm satisfies this definition in the third paragraph of the introduction [Our definition…].

  2. In a chaotic system, the orbits are not closed, but “randomly” jump from one point in phase space to another. As a consequence, any initial probability distribution spreads evenly in phase space. This effect is best exemplified by the phase space portraits of the kicked rotor - see for example https://commons.wikimedia.org/wiki/File:Kicked_Rotor_Phase_Portrait.png

  3. My discussion deals with random forces that average to zero and do not affect the stability of the fixed point. This is now clarified in the text: "A natural example is offered by the random forces associated with a coupling to a thermal bath."

  4. I thank the Referee for the suggestion and updated the title accordingly.

  5. Following the Referee’s suggestion, I now added a paragraph to the introduction, where I define the concept scale invariance in the usual sense, and relate it to my new work.

  6. Fixed

  7. I thank the Referee for spotting this severe typo, which was now fixed.

---

## Round 3 · Referee Report · Anonymous (Referee 3) · 2018-3-29

Strengths

The simple basic considerations presented in the paper are certainly to be recommended for publishing in SciPost. They concern a both striving and deep thematics of present-day research in the field of quantum dynamics. The paper is well readable and concise. Hence, I would like to recommend publication after the author has considered the following minor remarks and questions given below.

Weaknesses

The paper, in the present version, provides relatively little information on how the universality discussed is related to the universality commonly studied in the framework of the renormalisation group and of conformal theories. See points for amendmends below.

Report

The author studies a special type of universality in the dynamics of quantum systems which can be traced back to the unitarity of quantum mechanical evolution.

Starting with a simple symmetric quantum harmonic oscillator, he demonstrates that the marginal phase-space probability (phase-space probability integrated over momentum) of oscillator exhibits, in leading approximation, a single logarithmically divergent and thus scale invariant term around the stable fixed point of the classical motion, with a universal pre-factor. He then goes on showing that this remains true for interacting, i.e., non-linear oscillators in the vicinity of the fixed point, demonstrating this result for various integrable and non-integrable models. Extending the discussion, finally, to dissipative situations, the author can trace back the universal scaling behavior to the unitarity of the quantum evolution and thus to the underlying U(1) symmetry.
These findings allow him to classify generic perturbation terms to an action as “irrelevant”, “marginal”, and “relevant”, in analogy to renormalisation-group coupling flows near an RG fixed point.

Requested changes

  1. To what extent can one be sure that the presented scale invariance is “new” (see abstract)? Typically, when considering scale invariance in classical and quantum physics in the context of universality (mostly related to phase transitions), one considers correlation functions of different order as these are the observables in an experiment and for most non-trivial models can be computed in practice. Here, the author considers full phase-space distribution functions since he can compute them for the models of consideration. How does the universality showing up in these functions translate to universal behavior of a correlator?

The results presented show that the pointed-out universality occurs in the limit phi->0. In a field model, this corresponds typically to the mean-field approximation. Here this is confirmed, as in the scaling limit the effect of non-linearities essentially disappears.

  1. From that point of view, the results ask for being set into relation to typical RG statements concerning models in d dimensions, in particular depending on whether d is below or above the critical dimension, can be captured by mean-field theory or not. All the models studied here strictly speaking “live” in zero spatial dimensions. Also, the fixed points do not have anything to do with a phase transition per se. But the author uses the terminology of relevant and irrelevant couplings. Can the author say anything to this relation?

The author points out that at the (phase) transitions he studies with the different models, the universal scaling of the distribution function remains intact while the pre-factor changes in a non-analytic manner. To my apprehension, this can be traced back to the fact that for the models and parameters studied, the particular classical fixed point around which the log divergence occurs changes in the transition, as it is the case in symmetry breaking phase transitions. The scaling behavior, however, is independent of the criticality around these transitions, apart from a change in the pre-factor. So the universality pointed to rather seems to be related to the “trivial” Gaussian fixed point every non-linear model has at zero temperature and in the limit of vanishing non-linearity (this is briefly touched in the second par. of the introduction). These aspects should be commented on further, to not confuse the reader between common use of terminology and the one presented here.

  1. Remarks towards experiment:

From the discussion of dissipative motion I do not see yet how this can help an experimenter in practice. This should be made more specific, in case that the author wants to announce experimental considerations in the headline of Sect. 5.

  1. A few technical remarks:

  2. par. 2, first line: (x^2+p^2)/2

  3. Eq. 2: minus sign in lower integral limit missing, and factor 1/2 (though subleading) in argument of logarithm (also an “+O(x^2)” addition may help).
  4. Caption Fig. 1a: “vertical line”->”horizontal line”, and parentheses around (1/pi^2).
  5. p. 4, 2nd par.: Jacobi; and (1/pi^2)
  6. p. 5, 2nd par of 4., 1st line: “Eq.”->”Ref.”
  7. same par., last but 3 line: “The Dicke model … as the model defined in Eq. 3.”
  8. next par.: equals to Eq. 2->equals that in Eq. 2
  9. p. 6, last line: divergent

  • validity: top
  • significance: good
  • originality: high
  • clarity: high
  • formatting: excellent
  • grammar: excellent

Author:  Emanuele Dalla Torre  on 2018-07-18  [id 294]

(in reply to Report 3 on 2018-03-29)
Category:
answer to question

I wish to thank the Referee for carefully reviewing this manuscript and for her/his useful comments.

  1. The first comment by the Referee deals with the core novelty of this work. In the usual case, scale invariance is defined with respect to a rescaling of the space and time coordinates. In contrast, here I consider the rescaling of physical observables. This point is not better explained in the 3rd paragraph of the Introduction [“Our definition…”]. My scale invariance does not reveal itself as a universal correlation function, but rather as a universal distribution function.

The Referee correctly points out that my scaling procedure focuses on the regime of $\phi\to0$, where the systems is effectively linear. Nevertheless, this is not enough to allow us to disregard non-linearities. Indeed, for any finite $\phi_0\neq0$, there exists a time scale (inversely proportional to $\phi_0$) after which the effect of the non-linearity is significant. My analysis reveals that if the model is integrable, the universal scaling of the distribution function is conserved for asymptotically long times. In contrast, for ergodic systems, the non-linearities lead to thermalization and induce a flat the distribution probability. I now stress this point in the article:

"In this work, we show that the logarithmic divergence found in Eq. 3 is universal, as it is not affected by non-linearities. This result is non-trivial because, for any finite $x$, there exists a time after which the non-linearities have a significant effect. The logarithmic divergence is nevertheless preserved, as long as the stability of the $x=p=0$ point is preserved and the dynamics in its surroundings is characterized by invariant tori."

  1. The terminology of phase transitions is used to clarify that although some perturbations do not affect the scale invariance of the distribution functions (irrelevant), others do (relevant). As mentioned earlier, I was not able to establish a direct comparison between the (new) scale invariance of distribution functions and the (conventional) scale invariance of correlation functions of extended systems. In addition, one needs to consider that I am dealing with non-equilibrium processes, whose universality differs from the equilibrium one (see for example the case of the phase transition of asymmetric exclusion models in 1d)

The Referee correctly points out that although the logarithm is universal, its prefactor is not. Following her/his Referee’s suggestion, I rephrased a few sentences that referred to the logarithm, clarifying its non-universal nature. In addition, the Referee suggests to associate the observed scale invariance with a “Gaussian” fixed point because it is can be described using linear equations of motion. I agree with this identification and now say:

"In this sense, the present scale invariant states can be associated with a Gaussian fixed point. At equilibrium, these fixed points offer the simplest example of scale invariant critical points. An interesting question for further studies is whether distribution functions can show non-Gaussian fixed points that are scale invariant as a consequence of non-linear terms."

  1. Following the Referee’s suggestion, I have changed the name and the opening of the section on dissipation.

  2. I thank the Referee for spotting these typos, which were now fixed.

---

## Round 4 · Referee Report · Anonymous (Referee 3) · 2018-7-28

Strengths

see previous report

Weaknesses

see previous report and minor amendments

Report

The author amended his ms. according to the requests by the referees. Concerning his answers to my questions:

“1. The first comment by the Referee deals with the core novelty of this work. In the usual case, scale invariance is defined with respect to a rescaling of the space and time coordinates. In contrast, here I consider the rescaling of physical observables. This point is not better explained in the 3rd paragraph of the Introduction [“Our definition…”]. My scale invariance does not reveal itself as a universal correlation function, but rather as a universal distribution function.”

I do not think that there is a fundamental difference between the scaling considered by the author and conventional scaling: If one reduces the dimension to 0+1 and averages over time then the remaining space is zero-dimensional and scaling “only” affects observables. This is the same in a higher-dimensional system where operators also receive a scaling dimension which summarizes their scaling resulting from a rescaling of time (and spatial) arguments.

Also scaling of distribution functions as compared to correlation functions does not represent a difference as the distribution can be expanded and thus represented and generated by its moments which are the correlation functions. Hence, if all correlation functions scale, also the distribution function scales. The nature of the problems considered by the author suggests to consider distribution functions in the first place.

This is why I asked for the relation between the scaling of distributions and that of corr. functions in the paper. Maybe one could at least add a remark on that.

“The Referee correctly points… invariant tori.””

I agree.

I am also happy with answer and amendment concerning point 2.

Requested changes

Optional: see report

---

## Round 4 · Referee Report · Anonymous (Referee 1) · 2018-8-14

Strengths

see previous report

Weaknesses

see previous report

Report

I read through the author’s reply and through the relevant parts in the manuscript, and I can be brief, my comments are comprehensively addressed.
The notion of scale invariance put forward by the author is unconventional, only approximate, and one may have referenced what he means differently, but it is now clearly defined and no one will get confused based on that. So I would leave the responsibility for such choice to the author.

Requested changes

optional, see above

---

## Round 4 · Referee Report · Anonymous (Referee 2) · 2018-8-17

Report

I have read the new version of the manuscript by Dalla Torre entitled \emph{Scale invariant distribution functions in quantum systems with few degrees of freedom}, which has been resubmitted to SciPost.

The author has significantly changed the manuscript according to the referees comments. In particular the new version has now a more extended introduction and the appendix contains information on the dependence from the total spin S and the study of distribution function for different observables. The author has also answered convincingly to several issues raised by the Referees.

Overall, I think the manuscript contains interesting new results and deserves in this form publication in SciPost.

---

## Round 4 · Author Response

Dear Editor,
I sincerely thank all 3 Referees for carefully reading my manuscript and for their useful comments.
The Referees asked minor changes to the manuscript, which I implemented in the new version.
I think that the article should now be ready for publications.
Thank you for your time and consideration.
Best regards,
Emanuele Dalla Torre

---

## Round 4 · List of Changes

• Added a paragraph to the introduction, were I define the concept of scale invariance and explain the difference between the usual case and the new one.
  • Added two appendices on the effects of the finite size (A.3) and on the scaling of additional observables (A.4).
  • Added two footnotes on the formal definition of scale invariance (page 2) and Gaussian fixed points (page 3).
  • Improved the discussion on the effects of non-linearities (page 3), finite size (page 5), and random variables (page 7).

---

## Editorial Decision

published